# Optimal minimal residual disease threshold in pediatric acute myeloid leukemia: A retrospective cohort study based on the TARGET database

Xiong-yu Liao[1,2☯], Hong Zheng[1,2☯], Jian-pei Fang[1,2*], Dun-hua Zhou[1,2*], Kun-yin Qiu[1,2*]

1 Department of Hematology/Oncology, Children's Medical Center, Sun Yat-sen Memorial Hospital, Sun Yat-sen University, Guangzhou, P. R. China, 2 Guangdong Provincial Key Laboratory of Malignant Tumor Epigenetics and Gene Regulation, Sun Yat-sen Memorial Hospital, Sun Yat-sen University, Guangzhou, P. R. China

☯ These authors have contributed equally to this work as first authors.
* fangjpei@mail.sysu.edu.cn (J-pF); zhoudunh@mail.sysu.edu.cn (D-hZ); qiuky@mail2.sysu.edu.cn (K-yQ)

## Abstract

### Background

Minimal residual disease (MRD) monitoring is a cornerstone of risk stratification in pediatric acute myeloid leukemia (AML), with a threshold of 0.1% conventionally defining positivity by flow cytometry. Advances in flow cytometric technologies, enabling detection of leukemic cells with higher sensitivity and specificity, warrant a reevaluation of whether a lower threshold improves prognostic accuracy.

### Methods and findings

We conducted a retrospective cohort study using data from the Therapeutically Applicable Research to Generate Effective Treatments (TARGET)-AML initiative. The study population comprised 1,205 pediatric patients with de novo AML treated across Children's Oncology Group (COG) clinical trial centers. Patients were enrolled between September 1996 and December 2016, with a median follow-up of 6.2 years (range: 0.5–20.1 years). The primary objective was to compare the prognostic performance of the traditional MRD threshold (≥0.1%) with a lower threshold (≥0.05%) after induction courses 1 and 2. The main outcome measure was 5-year event-free survival (EFS). Analyses included Kaplan−Meier survival estimates, Cox proportional hazards models to calculate hazard ratios (HR) with 95% confidence intervals (CI), receiver operating characteristic (ROC) curves, and net reclassification improvement (NRI). The optimal threshold for predicting 5-year EFS, determined by ROC analysis, was 0.05% after both induction course 1 (AUC: 0.840, 95%CI[0.76,0.88]) and course 2 (AUC: 0.854, 95%CI[0.78,0.89]). The 0.05% threshold demonstrated higher HR for the first event than the 0.1% threshold (after course 1: HR=2.8, 95%CI[2.3,3.3];

**Data availability statement:** The data underlying this study are publicly available from the TARGET-AML database (accession number: TARGET-AML; URL: https://ocg.cancer.gov/programs/target/data-matrix). Researchers can access the data via the repository portal without restrictions. The custom R code used for statistical analyses is available as Supporting information (S1 Code). The study protocol was included as Supporting information (S1 Protocol).

**Funding:** This work was supported by Guangdong Basic and Applied Basic Research Foundation (NO.2024A1515012445 to ZDH, https://pro.gdstc.gd.gov.cn/egrantweb/), Guangdong Medical Scientific Research Foundation (A2024057 to QKY; B2025513 to LXY, https://pro.gdstc.gd.gov.cn/egrantweb/), Guangzhou Basic and Applied Basic Research Foundation (2024A04J4686 to QKY, https://gzsti.gzsi.gov.cn/pms/homepage.html), and Yat-sen Excellent Young Scientists Fund (2024A03J1185 to QKY, https://gzsti.gzsi.gov.cn/pms/homepage.html). The funders had no role in study design, data collection and analysis, decision to publish, or preparation of the manuscript.

**Competing interests:** The authors have declared that no competing interests exist.

**Abbreviations:** AIC, Akaike Information Criterion; AML, acute myeloid leukemia; AUC, area under the ROC curve; BM, bone marrow; CBF, core-binding factor; CI, confidence intervals; CNSL, central nervous system leukemia; COG, Children's Oncology Group; CR, complete remission; ELN, European LeukemiaNet; HR, hazard ratios; HR, high risk; IRB, Institutional Review Boards; LAIP, leukemia-associated immunophenotype; LR, low risk; MFC, multiparametric flow cytometry; MFC-MRD, multiparametric flow cytometry-based MRD; MRD, minimal residual disease; PB, peripheral blood; ROC, receiver operating characteristic; SCT, stem cell transplantation; SR, standard risk; STROBE, Strengthening the Reporting of Observational Studies in Epidemiology; TARGET, Therapeutically Applicable Research to Generate Effective Treatments; WBC, white blood cell.

$P < 0.001$; after course 2: $HR = 3.7$, $95\%CI[3.0, 4.6]$; $P < 0.001$). NRI analysis confirmed significant improvement in risk classification with the 0.05% threshold (overall NRI: 0.15 after course 1, 0.18 after course 2). The main limitation of this study is its retrospective design using historical data from trials conducted over 20 years, which may limit generalizability to contemporary treatments.

## Conclusions

A lower MRD threshold of 0.05% provides superior prognostic discrimination compared to the conventional 0.1% threshold in pediatric AML treated in previous COG trials. These findings support testing this more sensitive threshold in future clinical trial designs for improved risk-adapted therapy.

## Author summary

### Why was this study done?

- Doctors currently use a test called "minimal residual disease" (MRD) to help predict outcomes for children with a type of blood cancer called acute myeloid leukemia (AML), using a specific cutoff level (0.1%) to define a positive result.

- However, with advances in testing technology, it is unclear if this traditional cutoff is still the best one for identifying which children are at the highest risk of their cancer returning after treatment.

- This study was done to find out if using a lower, more sensitive cutoff (0.05%) could more accurately predict which children with AML are more likely to experience relapse or other treatment failures.

### What did the researchers do and find?

- We analyzed information from a large group of 1,205 children with AML treated in clinical trials to compare how well the traditional cutoff (0.1%) and the new, lower cutoff (0.05%) predicted their long-term outcomes.

- We found that the lower cutoff of 0.05% was better at separating children into groups with different risks. Children with MRD levels at or above 0.05% had a nearly 3 times higher risk of relapse or other events after the first round of chemotherapy, and a risk almost 4 times higher after the second round, compared to those with lower levels.

- The analysis showed that using the 0.05% cutoff improved the accuracy of risk classification for patients by 15%–18% compared to the traditional 0.1% cutoff.

**What do these findings mean?**

- These results suggest that lowering the MRD cutoff to 0.05% could help doctors make more precise decisions about which children need stronger or different treatments to prevent their cancer from coming back.

- This finding may lead to changes in how future clinical trials for childhood AML are designed, using this more sensitive threshold to guide therapy.

- A main limitation of the study is that it looked back at historical data from trials conducted over 20 years, rather than testing the new cutoff in a new prospective study, which means the findings should be confirmed in future research.

## 1. Introduction

Pediatric acute myeloid leukemia (AML) represents a heterogeneous group of hematologic malignancies characterized by the clonal proliferation of myeloid precursors with impaired differentiation [1]. Despite significant advancements in risk stratification and treatment intensification, including allogeneic hematopoietic stem cell transplantation (SCT) for high-risk patients, relapse remains a leading cause of mortality, with event-free survival (EFS) rates plateauing in many contemporary trials [2–4]. This underscores the critical need for refined and highly sensitive prognostic biomarkers to better identify patients at the highest risk of treatment failure who may benefit from early intervention or alternative therapeutic strategies.

The detection of minimal residual disease (MRD) has emerged as a cornerstone of risk-adapted therapy in acute leukemias [5]. Among the detection methods for MRD in AML, multiparametric flow cytometry-based MRD (MFC-MRD) has the lowest sensitivity. However, it offers the fastest detection speed, is the most widely used, and is applicable to the largest patient population [6]. In pediatric AML, the presence of MFC-MRD following induction chemotherapy is among the most powerful independent predictors of relapse and survival [7]. Conventionally, a threshold of 0.1% leukemic cells within the bone marrow (BM), has been widely adopted to define MFC-MRD positivity and guide clinical decisions, and this threshold has proven valuable in discriminating outcomes in numerous studies [5,8–11]. However, with the continuous evolution of flow cytometric technologies, offering ever-increasing sensitivity and specificity, the question arises whether this traditional threshold remains optimal. There is a growing body of speculative evidence suggesting that a lower, more sensitive threshold might identify a subset of patients with ultra-low-level residual disease who still harbor a significantly elevated relapse risk compared to those in a deeper molecular remission [10,12,13].

Despite this compelling rationale, the validation of a novel, lower MRD threshold in a large, well-annotated pediatric AML cohort has been lacking. Key questions remain unanswered: Does a threshold lower than 0.1% provide superior prognostic discrimination after initial (Course 1) and subsequent (Course 2) induction therapy? Does it improve risk classification across established genetic risk groups? Furthermore, does it offer a net reclassification improvement (NRI) over the current standard, justifying a change in clinical practice?

To address these pivotal questions, we leveraged data from the comprehensive Therapeutically Applicable Research to Generate Effective Treatments (TARGET) AML initiative. This study aimed to: (1) compare the prognostic performance of the 0.1% versus 0.05% MRD thresholds for 5-year EFS using ROC and survival analyses; (2) quantify improvement via NRI; and (3) validate the threshold across genetic risk subgroups. Our findings advocate for a paradigm shift in the definition of MRD negativity in pediatric AML, proposing that adopting a 0.05% threshold enhances prognostic accuracy, improves risk stratification, and could ultimately guide more precise and effective risk-adapted therapies.

## 2. Patients and methods

### 2.1 Study population and data source

This retrospective cohort study utilized data from the TARGET-AML initiative, a collaborative project by the National Cancer Institute. The analysis included pediatric de novo-AML patients (aged <18 years at diagnosis) with available

MFC-MRD data after induction therapy. Comprehensive baseline clinical and molecular characteristics were collected, including age, white blood cell (WBC) count at diagnosis, peripheral blood (PB) and BM blast percentages, central nervous system leukemia (CNSL) status, and molecular abnormalities (CEBPA, FLT3-ITD, NPM1, and WT1 mutations). Detailed baseline characteristics of the included participants are presented in Table 1. The original data collection within the Children's Oncology Group (COG) trials was approved by the Institutional Review Boards (IRB) of all participating centers, including the lead IRB at the National Cancer Institute, with written informed consent obtained from all participants or their guardians [2]. The analysis of de-identified data for this specific study was reviewed and approved by the IRB of Sun Yat-sen Memorial Hospital (Approval No. SYSEC2-2025-BA-885). This study is reported as per the Strengthening the Reporting of Observational Studies in Epidemiology (STROBE) guideline (S1 Checklist). For this specific retrospective analysis of the de-identified data from the TARGET database, the IRB of Sun Yat-sen Memorial Hospital granted a waiver of informed consent. Our analysis met the criteria for waiver as it involved no more than minimal risk and could not practicably be carried out without the waiver.

## 2.2  Treatment protocols and response assessment

Patients were treated under three protocols AAML03P1, AAML0531 and AAML1031, which typically included anthracycline and cytarabine-based induction chemotherapy. Patients were classified into risk groups—low-risk (LR), standard-risk (SR), and high-risk (HR)—according to the contemporary 2025 European LeukemiaNet (ELN) recommendations for pediatric AML [12]. The HR group was defined by the presence of adverse genetic features, including but not limited to: FLT3-ITD with high allelic ratio, RUNX1 mutations, ASXL1 mutations, TP53 mutations, MECOM rearrangements, complex karyotype, monosomal karyotype (e.g., monosomy 5/del(5q), monosomy 7), and certain KMT2A-rearranged subtypes. The LR group included core-binding factor (CBF) AML[t(8;21) and inv(16)/t(16;16)] and cases with biallelic CEBPA or NPM1 mutations. All other cases were classified as SR. HR patients in first complete remission (CR) underwent SCT. Response to therapy was assessed after Induction Course 1 and Course 2. The timing for initiating Induction Course 2 was defined by the respective COG trial protocols, based primarily on hematologic recovery and not on MRD levels measured after Course 1. CR was defined according to standard international criteria, which include <5% blasts in a morphologically normal BM aspirate with recovery of PB counts [8].

## 2.3  MRD assessment

MRD was quantified using multiparameter flow cytometry (MFC) with a minimum of 500,000 events acquired, following the standardized antibody panels and leukemia-associated immunophenotype (LAIP) approaches detailed in the COG reference laboratory protocols [10]. In cases where a definitive LAIP was not identifiable, a 'different-from-normal' approach was employed as per standardized guidelines. Sensitivity was maintained at 0.01% throughout the study period. BM aspirates were collected at the end of Induction Course 1 and Induction Course 2. The MRD level was expressed as a percentage of total nucleated cells. For the primary analysis, MRD status was evaluated using two distinct thresholds: the traditional threshold of ≥0.1% and the novel proposed threshold of ≥0.05%. Further, for dose–response analysis, MRD levels were categorized into three ordered groups: <0.05%, 0.05% to <0.1%, and ≥0.1%.

## 2.4  Hypotheses and analysis plan

To ensure transparency and reproducibility, we pre-specified our analytical approach prior to conducting data analysis. This section details the study's hypotheses, the planned methods to test them, the analyses ultimately performed, and explanations for any deviations from the original plan.

### 2.4.1  Pre-specified hypotheses.  We aimed to test the following primary hypotheses:

**Table 1. Baseline characteristics stratified by first event status. Between-group differences were tested using $\chi^2$ tests for categorical variables and Mann–Whitney $U$ tests for continuous variables; all $p$-values are two-sided.**

| Characteristics | Total ($n$ = 1,205) | First Event Status | | $P$-value |
|---|---|---|---|---|
| | | Non-event groups ($n$ = 594) | Event group ($n$ = 611) | |
| Sex | | | | 0.072 |
| Male | 624 (51.8%) | 292 (49.2%) | 332 (54.3%) | |
| Female | 581 (48.2%) | 302 (50.8%) | 279 (45.7%) | |
| Age group | | | | 0.966 |
| <10 | 656 (54.4%) | 323 (54.4%) | 333 (54.5%) | |
| ≥10 | 549 (45.6%) | 271 (45.6%) | 278 (45.5%) | |
| Chemotherapy Protocol | | | | 0.580 |
| AAML1031 | 669 (55.5%) | 334 (56.2%) | 335 (54.8%) | |
| AAML0531 | 473 (39.3%) | 226 (38.0%) | 247 (40.4%) | |
| AAML03P1 | 63 (5.2%) | 34 (5.7%) | 29 (4.7%) | |
| WBC | 25.9 (0.6–918.5) | 20.6 (0.6–549.9) | 35.1 (0.7–918.5) | <0.001 |
| BM blast(%) | 70.0 (0.0—00.0) | 68.0 (0.0–100.0) | 73.0 (0.0–100.0) | <0.001 |
| Peripheral blasts (%) | 42.0 (0.0–99.0) | 38.0 (0.0–98.0) | 46.0 (0.0–99.0) | 0.005 |
| Risk group | | | | <0.001 |
| Low-risk | 483 (40.8%) | 327 (56.4%) | 156 (25.8%) | |
| Standard-risk | 574 (48.4%) | 202 (34.8%) | 372 (61.5%) | |
| High-risk | 128 (10.8%) | 51 (8.8%) | 77 (12.7%) | |
| Karyotype | | | | <0.001 |
| KMT2A | 256 (21.7%) | 102 (17.6%) | 154 (25.6%) | |
| t(8;21) | 183 (15.5%) | 129 (22.3%) | 54 (9.0%) | |
| inv(16) | 146 (12.4%) | 91 (15.7%) | 55 (9.1%) | |
| Normal | 279 (23.6%) | 141 (24.4%) | 138 (22.9%) | |
| Other | 317 (26.8%) | 116 (20.0%) | 201 (33.4%) | |
| FLT3-ITD Status | | | | 0.028 |
| FLT3-ITD-wt | 1,021 (84.7%) | 517 (87.0%) | 504 (82.5%) | |
| FLT3-ITD-mt | 184 (15.3%) | 77 (13.0%) | 107 (17.5%) | |
| NPM1 Status | | | | <0.001 |
| NPM1-wt | 1,086 (90.4%) | 508 (86.0%) | 578 (94.8%) | |
| NPM1-mt | 115 (9.6%) | 83 (14.0%) | 32 (5.2%) | |
| CEBPA Status | | | | 0.016 |
| CEBPA-wt | 1,124 (93.7%) | 543 (92.0%) | 581 (95.4%) | |
| CEBPA-mt | 75 (6.3%) | 47 (8.0%) | 28 (4.6%) | |
| WT1 Status | | | | 0.006 |
| WT1-wt | 492 (92.5%) | 246 (95.7%) | 246 (89.5%) | |
| WT1-mt | 40 (7.5%) | 11 (4.3%) | 29 (10.5%) | |
| FAB Category | | | | <0.001 |
| M0 | 37 (4.1%) | 5 (1.1%) | 32 (7.0%) | |
| M1 | 119 (13.3%) | 54 (12.4%) | 65 (14.3%) | |
| M2 | 265 (29.7%) | 161 (36.8%) | 104 (22.9%) | |
| M4 | 206 (23.1%) | 104 (23.8%) | 102 (22.4%) | |
| M5 | 199 (22.3%) | 80 (18.3%) | 119 (26.2%) | |
| M6 | 10 (1.1%) | 4 (0.9%) | 6 (1.3%) | |
| M7 | 56 (6.3%) | 29 (6.6%) | 27 (5.9%) | |

*(Continued)*

**Table 1.** (Continued)

| Characteristics | Total (*n* = 1,205) | First Event Status | | P-value |
|---|---|---|---|---|
| | | Non-event groups (*n* = 594) | Event group (*n* = 611) | |
| CNS disease | | | | 0.046 |
| CNS1 | 451 (69.7%) | 237 (73.4%) | 214 (66.0%) | |
| CNS2 | 145 (22.4%) | 68 (21.1%) | 77 (23.8%) | |
| CNS3 | 51 (7.9%) | 18 (5.6%) | 33 (10.2%) | |
| Chloroma | | | | 0.233 |
| No | 754 (89.2%) | 373 (90.5%) | 381 (88.0%) | |
| Yes | 91 (10.8%) | 39 (9.5%) | 52 (12.0%) | |
| MRD at end of Course 1 | | | | <0.001 |
| <0.1% | 924 (76.7%) | 525 (88.4%) | 399 (65.3%) | |
| ≥0.1% | 281 (23.3%) | 69 (11.6%) | 212 (34.7%) | |
| MRD at end of Course 2 | | | | <0.001 |
| <0.1% | 1,073 (89.0%) | 571 (96.1%) | 502 (82.2%) | |
| ≥0.1% | 132 (11.0%) | 23 (3.9%) | 109 (17.8%) | |
| SCT in 1st CR | | | | 0.339 |
| No | 943 (84.1%) | 503 (85.1%) | 440 (83.0%) | |
| Yes | 178 (15.9%) | 88 (14.9%) | 90 (17.0%) | |

**Abbreviations:** HL, Hyperleukopenia; WBC, white blood cell counts; PB blast, peripheral blood blast; BM, bone marrow blast; CNSL, central nervous system leukemia; CEBPA, CCAAT/enhancer binding protein alpha; FLT3-ITD, fms-related tyrosine kinase 3; NPM1, nucleophosmin 1; WT1, wilms tumor 1; MRD, minimal residual disease; SCT, stem cell transplantation.

$H_1$: A minimal residual disease (MRD) threshold of ≥0.05% after induction therapy would provide superior prognostic discrimination for 5-year EFS compared to the conventional threshold of ≥0.1%.

$H_2$: The use of the 0.05% MRD threshold would result in a significant NRI over the 0.1% threshold.

$H_3$: The prognostic value of the 0.05% MRD threshold would be consistent across predefined genetic risk subgroups (low-, standard-, and high-risk).

**2.4.2 Planned analytical methods.** The analysis plan, documented before data interrogation, outlined the following methods to test these hypotheses:

For $H_1$ and $H_2$: Receiver operating characteristic (ROC) curve analysis with Youden's index would be used to determine the optimal prognostic threshold. The prognostic discrimination of the two thresholds would be compared using Kaplan–Meier survival estimates with log-rank tests, Cox proportional hazards regression to calculate hazard ratios (*HR*) with 95% confidence intervals (*CI*), and categorical NRI analysis.

For $H_3$: Subgroup analyses using stratified Cox regression models were planned to assess the consistency of the MRD threshold's effect within each genetic risk group.

Multivariable Model Building: The initial plan specified entering variables with a univariate *p*-value <0.1 into the multivariable Cox regression model.

**2.4.3 Analyses performed.** The analyses conducted align closely with the pre-specified plan. We performed:

• ROC analysis for 5-year EFS after induction courses 1 and 2.

• Univariate and multivariable Cox regression analyses for EFS.

- Kaplan–Meier survival analysis with log-rank tests.

- Categorical NRI analysis.

- Pre-planned subgroup analyses within genetic risk categories.

**2.4.4 Deviations from the planned analysis.** Modifications were made to the initial analysis plan during the peer review process to enhance methodological rigor. These changes were implemented in direct response to reviewer feedback and are detailed below with transparent rationale.

Variable Selection Method: The initial plan to use a univariate *P*-value threshold (<0.1) for variable selection in the multivariable model was replaced with the Akaike Information Criterion (AIC) for model selection. This change was made to optimize prediction performance and is a statistically superior approach, as noted by Reviewer. It does not affect the interpretation of the primary findings related to the MRD threshold.

No other data-driven or unplanned exploratory analyses were performed as part of the primary results reported in this manuscript.

## 2.5 Statistical analysis

All statistical analyses were performed using R software (version 4.1.0) and SPSS (version 26.0). EFS was defined as the time from diagnosis to the first event (induction failure, relapse, secondary malignancy, or death from any cause) or the date of last follow-up for censored observations. The association between variables and first event was assessed using *HR* with 95%*CI*.

**2.5.1 Univariate and multivariable analysis.** The association between clinical variables, including both MRD thresholds and EFS was first assessed using univariate Cox proportional hazards models. Variable selection for the multivariable Cox proportional hazards model was performed using the AIC to optimize prognostic prediction performance for the first event. Results were reported as *HR* with 95%*CI*.

**2.5.2 Survival curve estimation.** Kaplan–Meier survival curves were generated to visualize the EFS probabilities stratified by MRD status (using both thresholds) after each induction course. The statistical significance of the differences between survival curves was compared using the log-rank test.

**2.5.3 ROC curve analysis.** ROC curve analysis was employed to evaluate the predictive accuracy of continuous MRD levels (after Course 1 and Course 2) for 5-year EFS status and to determine the optimal prognostic threshold value. The optimal threshold was selected based on Youden's index (J). In recognition of the greater clinical consequence of false negatives (missed relapses), a supplementary sensitivity analysis was performed. In this analysis, sensitivity was weighted twice as heavily as specificity to identify a clinically optimized threshold.

**2.5.4 NRI.** To quantify the improvement in risk prediction offered by the novel MFC-MRD threshold (0.05%) compared to the traditional threshold (0.1%), a categorical NRI analysis was performed for EFS. This analysis measures the correct movement of individuals across risk categories (e.g., from LR to HR for those who experienced an event, and from HR to LR for those who did not). The NRI was calculated to quantify the improvement in risk prediction when replacing the traditional threshold (0.1%) with the novel threshold (0.05%) within the final multivariable prognostic model.

**2.5.5 Subgroup analysis.** To assess the consistency of the prognostic impact of the novel MRD threshold across different risk groups, subgroup analyses were conducted within patients classified as LR, SR, and HR based on established cytogenetic and molecular criteria [14]. A two-sided *p*-value of <0.05 was considered statistically significant for all tests, except where specified for variable selection in multivariable modeling.

Totally, this study was a retrospective analysis of an existing cohort. The core analytical plan—specifically, the comparison of the prognostic performance of the 0.05% MRD threshold versus the conventional 0.1% threshold for 5-year EFS using Kaplan–Meier analysis, Cox regression, and NRI—was defined prior to conducting the data analysis for this specific research question. This plan was documented in the study protocol that was submitted for IRB approval.

# 3. Results

## 3.1 Study participants

Demographic, clinical, and laboratory data for children under 18 years of age with de novo-AML were obtained from the TARGET database (version dated April 28, 2021) (Table 1). A total of 1,205 pediatric patients with non-M3 AML were initially identified from September 1996 to December 2016. Among them, there were 549 pediatric patients (45.6%) aged over 10 years, and 624 male patients (51.8%). Analysis of 1,205 pediatric AML patients revealed significant differences in baseline characteristics between those with ($n = 611$) and without ($n = 594$) events. The event group presented with significantly higher initial disease burden, including WBC count (76.7 versus $53.8 \times 10^9$/L, $P < 0.001$) and BM blast percentage (66.7% versus 62.3%, $P = 0.002$). A pronounced imbalance was observed in genetic risk groups ($P < 0.001$). The non-event group had a higher proportion of low-risk patients (56.4% versus 25.8%), while the event group was enriched with standard-risk patients (61.5% versus 34.8%). The event group had a higher frequency of adverse molecular markers, including FLT3-ITD (17.5% versus 13.0%, $P = 0.028$) and WT1 mutations (10.5% versus 4.3%, $P = 0.006$), but a lower frequency of favorable NPM1 mutations (5.2% versus 14.0%, $P < 0.001$). The event group had lower CR rates after Course 1 (74.2% versus 86.3%, $P < 0.001$) and significantly higher levels of MRD at both Course 1 (5.6% versus 0.5%, $P < 0.001$) and Course 2 (3.6% versus 0.1%, $P < 0.001$) when using the traditional 0.1% threshold.

## 3.2 MRD levels after induction therapy and their association with prognosis

Following Induction Course 1, the median MRD level among patients was 0.0% (range: 0%–92%). After Induction Course 2, the median MRD level was also 0.0% (range: 0%–75%). Using the 0.1% threshold, 23.3% (281/1205) of patients were MRD-positive after Course 1 and 11.0% (132/1205) after Course 2. The majority of patients (76.7% after Course 1 and 89% after Course 2) had MRD levels below the conventional 0.1% threshold, underscoring the potential clinical utility of a more sensitive threshold for risk stratification in this large subgroup. Applying the 0.05% threshold, the proportion of MRD-positive patients increased to 30.1% after Course 1 and 15.5% after Course 2. Using the conventional MRD threshold of 0.1%, patients with MRD < 0.1% after Induction Course 1 ($n = 924$) exhibited a significantly superior 5-year EFS rate compared to those with MRD ≥ 0.1% ($n = 281$) (56.5%,95%$CI$[53.4%,59.8%] versus 24.4%,95%$CI$[19.8%,30.1%]; $P < 0.001$; Fig 1A). A similar advantage was observed after Induction Course 2, where the 5-year EFS rate was significantly higher in the MRD < 0.1% group ($n = 1,068$) than in the MRD ≥ 0.1% group ($n = 137$) (53.2%,95%$CI$[50.1%,56.1%] versus 17.2%, 95%$CI$[11.7%,24.9%]; $P < 0.001$; Fig 1B). Univariate Cox regression analysis confirmed that MRD ≥ 0.1% following Induction Course 1 was significantly associated with first event ($HR = 1.95$, 95%$CI$[1.1,3.2]; $P = 0.027$). Consistently, MRD ≥ 0.1% after Induction Course 2 was also a predictor of first event ($HR = 2.7$, 95%$CI$[2.3,3.2]; $P < 0.001$).

## 3.3 Determination of the optimal prognostic threshold for MRD at the end of induction therapy

To identify the optimal threshold of post-induction MRD for predicting 5-year first event, ROC curve analysis was performed separately at the end of Induction Course 1 and Induction Course 2. At the end of Induction Course 1, the area under the ROC curve (AUC) was 0.840 (95%$CI$[0.76,0.88], indicating that MRD level after Induction Course 1 possesses good prognostic value. The optimal threshold value, determined by the maximum Youden's index, was 0.05%, which corresponded to a sensitivity of 78.9% and a specificity of 83.9% (Fig 2A). Similarly, at the end of Induction Course 2, the AUC of the ROC curve was 0.854 (95%$CI$[0.78,0.89], demonstrating that MRD level after Induction Course 2 also exhibits good prognostic value. The optimal threshold value identified by the maximum Youden's index was likewise 0.05%, associated with a sensitivity of 80.2% and a specificity of 81.9% (Fig 2B). The point of maximum Youden's index (J) for both time points consistently corresponded to a threshold value of 0.05%, establishing it as the optimal threshold for prognostic prediction. The sensitivity analysis, which prioritized the minimization of false negatives by weighting sensitivity twice as

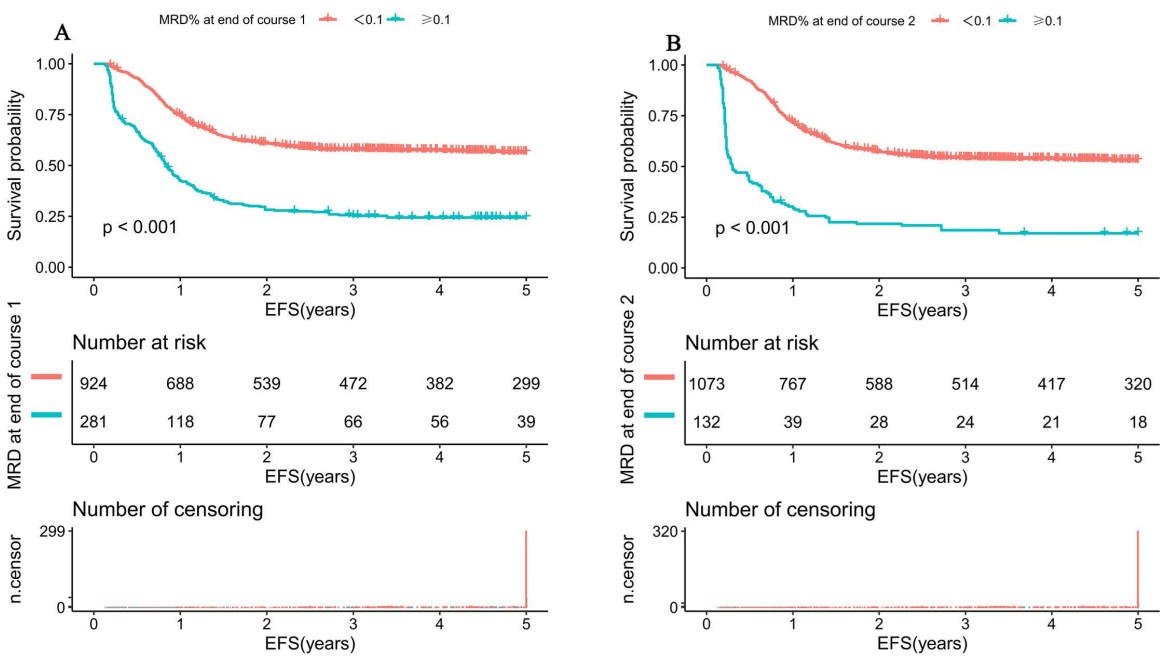

**Fig 1. Kaplan–Meier estimates of Event-Free Survival (EFS) stratified by traditional MRD threshold of 0.1% after induction therapy in pediatric AML.** Survival curves were compared using the log-rank test; *p*-values are two-sided. **(A)** EFS according to MRD status after the end of induction Course 1 by using the threshold of 0.1%. **(B)** EFS according to MRD status after the end of induction Course 2 by using the threshold of 0.1%.

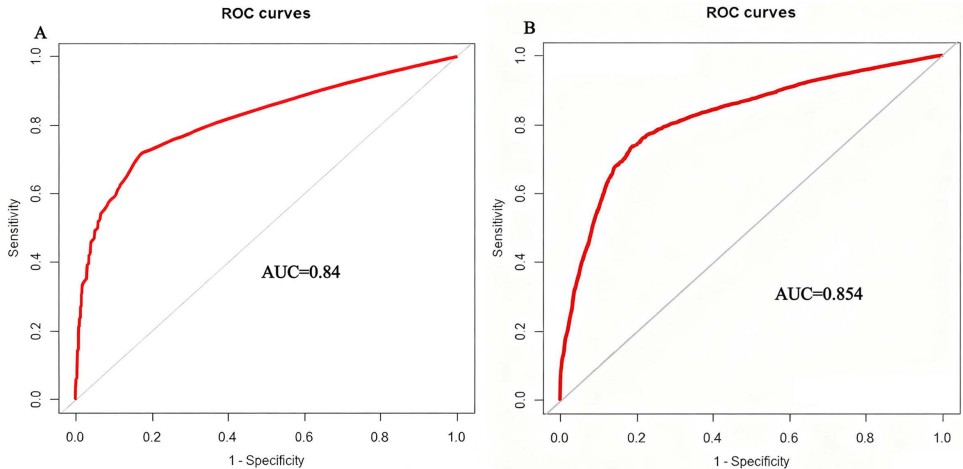

**Fig 2. Receiver operating characteristic (ROC) curve analysis for Minimal residual disease (MRD) levels in predicting 5-year event-free survival (EFS) status in pediatric AML.** Area under the curve (AUC) differences were tested using DeLong's method; *p*-values are two-sided. **(A)** ROC analysis for MRD measured after the end of Course 1. **(B)** ROC analysis for MRD measured after the end of Course 2.

heavily as specificity, confirmed that a threshold of 0.05% remained optimal, supporting the robustness of this cut-off value for clinical prognosis. In addition, for the conventional 0.1% threshold, the sensitivity and specificity for predicting 5-year EFS were 70.1% and 80.5% after Course 1, and 72.3% and 80.1% after Course 2, respectively.

### 3.4 Comparison of survival analyses based on the novel threshold for MRD

We further compared survival outcomes using the novel threshold for MRD (0.05%) versus the traditional threshold (0.1%). Kaplan–Meier analysis revealed that when patients were stratified by the 0.05% threshold after Induction Course 1 (MRD<0.05% versus ≥0.05%), the difference in 5-year EFS between the two groups was more pronounced (57.5%, 95%CI[54.4%,60.9%] versus 23.4%, 95%CI[18.9%,28.9%], P<0.001; Fig 3A), with a higher hazard ratio

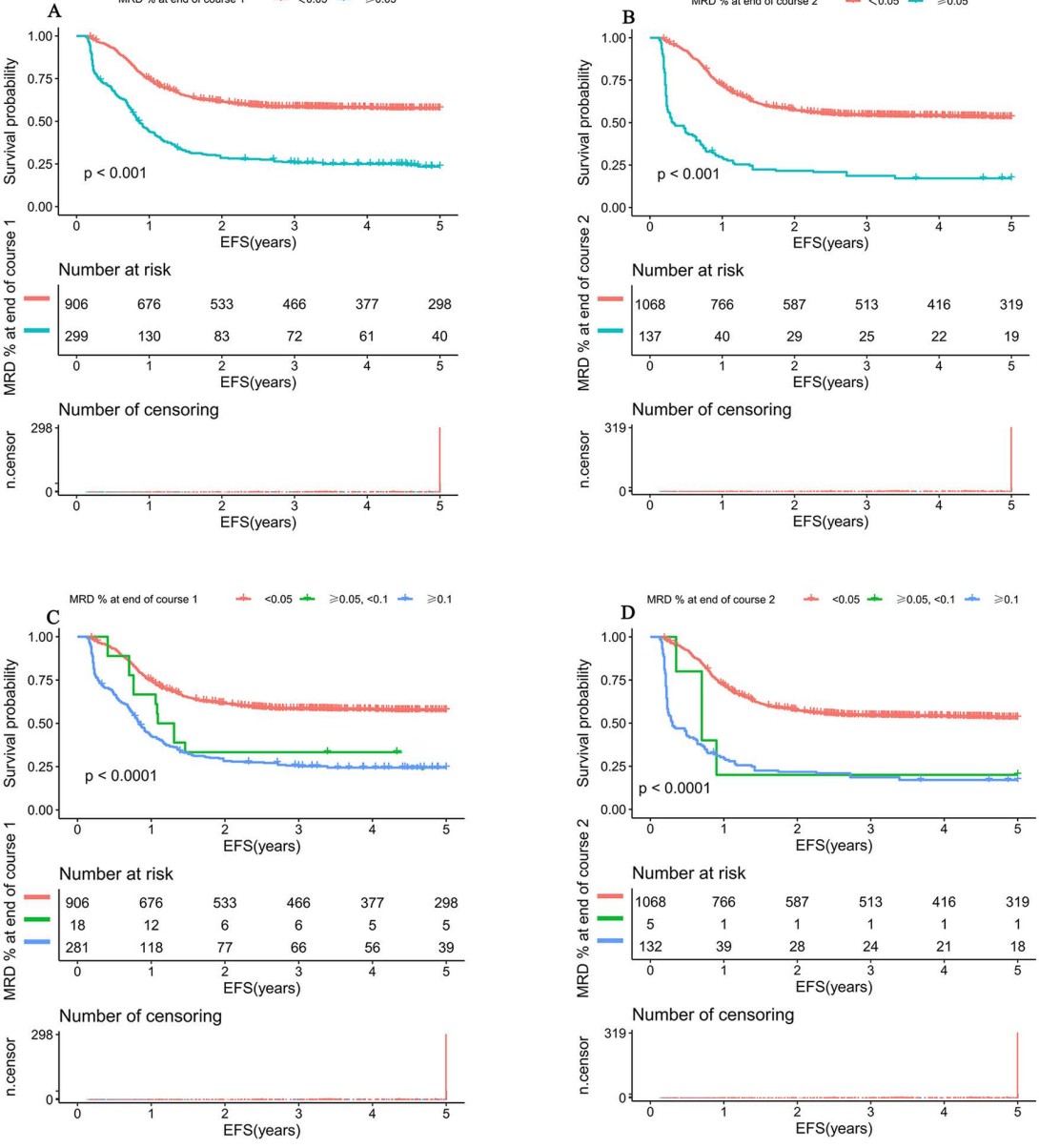

**Fig 3. Kaplan–Meier estimates of event-free survival (EFS) stratified by the different MRD threshold after induction therapy in pediatric AML.** Survival curves were compared using the log-rank test; p-values are two-sided. **(A)** EFS according to MRD status after the end of induction Course 1 by using the new threshold of 0.05%. **(B)** EFS according to MRD status after the end of induction Course 2 by using the new threshold of 0.05%. **(C)** EFS according to MRD status after the end of induction Course 2 by using the threshold of 0.05% and 0.1%. **(D)** EFS according to MRD status after the end of induction Course 2 by using the threshold of 0.05% and 0.1%.

(*HR*=2.8,95%*CI*[2.3,3.3]; *P* < 0.001) than that obtained using the 0.1% threshold (*HR*=1.9, 95%*CI*[1.1, 3.2]; *P*=0.027) (Table 2). Similarly, when patients were stratified by the 0.05% threshold after Induction Course 2 (MRD<0.05% versus≥0.05%), the difference in 5-year EFS remained more significant (53.2%,95%*CI*[50.2%,56.3%]; versus 17.2%, 95%*CI*[11.9%,24.9%]; *P*<0.001; Fig 3B), and the corresponding hazard ratio (*HR*=3.7, 95%*CI*[3.0,4.6] was also greater than that achieved with the 0.1% threshold (*HR*=2.7,95%*CI*[2.3,3.2]; *P*<0.001) (Table 2). Moreover, using the three-category variable, a clear dose–response relationship was observed after Induction Course 1: compared to patients with MRD<0.05% (EFS:57.5%, 95%*CI*[54.4%,60.9%], those with MRD 0.05% to <0.1% had an EFS of 33.3% (95%*CI*[17.1%,64.1%], and patients with MRD≥0.1% had an EFS of 24.4% (95%*CI*[19.8%,30.7%]; *P*<0.001) (Fig 3C). Similar results were also observed after the Induction Course 2 (Fig 3D).

### 3.5  NRI analysis

To quantify the improvement in predictive performance offered by the 0.05% MRD threshold within a comprehensive prognostic framework, we performed NRI analysis based on the multivariable Cox model, NRI was assessed following induction Course 1 and Course 2. After induction Course 1, the overall NRI was 0.15 (95%*CI*[0.05,0.25]; *P*=0.003). This improvement was driven by an event-group NRI of 0.10, reflecting better identification of patients who experienced recurrence, and a non-event-group NRI of 0.05, indicating more accurate classification of relapse-free patients (Table 3). Similarly, after induction Course 2, the overall NRI increased to 0.18 (95%*CI*[0.08,0.28]; *P*=0.003), with an event-group NRI of 0.12 and a non-event-group NRI of 0.06 (Table 4). These results consistently demonstrate that the 0.05% MRD threshold offers statistically significant and clinically relevant improvement in risk stratification over the 0.1% threshold at both post-induction time points.

### 3.6  Multivariable analysis

To validate whether MRD≥0.05% was an independent prognostic factor, we performed a multivariable Cox regression analysis. The results demonstrated that WT1 mutation, SCT in 1st CR, MRD≥0.1%, and MRD≥0.05% were all independent risk factors for adverse EFS. Notably, MRD≥0.05% emerged as the strongest predictor (Course 1: adjusted *HR*=6.7, 95%*CI*[1.8,24]; *P*=0.004; Course 2: adjusted *HR*=3.3, 95%*CI*[1.3,10]; *P*=0.011) (Table 5).

### 3.7  Validation of the novel threshold across risk subgroups

The prognostic validity of the 0.05% threshold was consistent across all predefined genetic risk groups—LR, SR, and HR—as detailed in Fig 4. Within each defined risk stratum, patients exhibiting MRD levels ≥0.05%—whether assessed after induction Course 1 or Course 2—demonstrated statistically significant inferior EFS compared to those with MRD levels below 0.05%. This consistent pattern was observed across LR, SR, and HR subgroups, underscoring the robust and independent prognostic value of the 0.05% threshold irrespective of the timing of assessment during induction therapy. The association remained significant even after adjusting for baseline molecular and clinical covariates, highlighting the critical clinical relevance of this lower MRD threshold for risk stratification and potential therapeutic decision-making.

## 4.  Discussion

The majority of patients in the present study had MRD levels below the conventional 0.1% threshold, underscoring the potential clinical utility of a more sensitive threshold for risk stratification in this large subgroup. This comprehensive reevaluation of the optimal MRD threshold in pediatric AML, utilizing data from the large, well-annotated TARGET cohort, demonstrates that a lower threshold of 0.05% provides superior prognostic discrimination compared to the conventional 0.1% threshold. The optimal MRD threshold of 0.05% was primarily identified using Youden's index, a standard metric that maximizes overall discriminative ability by assigning equal weight to sensitivity and specificity. We acknowledge that in the clinical context of AML, failing to identify a patient at high risk of relapse (a false negative) is typically considered more

# PLOS Medicine

**Table 2. Univariate and analysis for first event among pediatric AML. *P*-values were derived from Wald tests; all *p*-values are two-sided.**

| Variables | HR(95%CI) | P value |
|---|---|---|
| Sex | | |
| Male | Ref. | |
| Female | 0.8 (0.7, 1.0) | 0.021 |
| Age group | | |
| <10 | Ref. | |
| ≥10 | 1.0 (0.8, 1.1) | 0.809 |
| Chemotherapy Protocol | | |
| AAML1031 | Ref. | |
| AAML0531 | 1.0 (0.9, 1.2) | 0.602 |
| AAML03P1 | 0.8 (0.5, 1.2) | 0.217 |
| Risk group | | |
| Low-risk | Ref. | |
| Standard-risk | 2.8 (2.3, 3.3) | <0.001 |
| High-risk | 2.7 (2.0, 3.5) | <0.001 |
| WBC | | |
| $<50 \times 10^9$/L | Ref. | |
| $≥50 \times 10^9$/L | 1.4 (1.2, 1.7) | <0.001 |
| BM blast(%) | 1.0 (1.0, 1.0) | 0.003 |
| Peripheral blasts (%) | 1.0 (1.0, 1.0) | 0.006 |
| Karyotype | | |
| KMT2A | Ref. | |
| t(8;21) | 0.4 (0.3, 0.5) | <0.001 |
| inv(16) | 0.5 (0.4, 0.7) | <0.001 |
| Normal | 0.8 (0.6, 1.0) | 0.024 |
| Other | 1.1 (0.9, 1.3) | 0.451 |
| FLT3-ITD Status | | |
| FLT3-ITD-wt | Ref. | |
| FLT3-ITD-mt | 1.4 (1.1, 1.7) | 0.002 |
| NPM1 Status | | |
| NPM1-wt | Ref. | |
| NPM1-mt | 0.4 (0.3, 0.6) | <0.001 |
| CEBPA Status | | |
| CEBPA-wt | Ref. | |
| CEBPA-mt | 0.6 (0.4, 0.9) | 0.011 |
| WT1 Status | | |
| WT1-wt | Ref. | |
| WT1-mt | 2.1 (1.4, 3.0) | 0.001 |
| FAB Category | | |
| M0 | Ref. | |
| M1 | 0.4 (0.3, 0.7) | 0.001 |
| M2 | 0.3 (0.2, 0.4) | 0.001 |
| M4 | 0.4 (0.3, 0.6) | 0.001 |
| M5 | 0.5 (0.4, 0.8) | 0.002 |
| M6 | 0.6 (0.2, 1.3) | 0.181 |
| M7 | 0.4 (0.2, 0.7) | <0.001 |

*(Continued)*

**Table 2.** (Continued)

| Variables | HR(95%CI) | P value |
|---|---|---|
| Chloroma | | |
| No | Ref. | |
| Yes | 1.2 (0.9, 1.6) | 0.164 |
| MRD% at end of Course 1 | | |
| <0.1 | Ref. | |
| ≥0.1 | 1.9 (1.1, 3.2) | 0.027 |
| MRD% at end of Course 2 | | |
| <0.1 | Ref. | |
| ≥0.1 | 2.7 (2.3, 3.2) | <0.001 |
| SCT in 1st CR | | |
| No | Ref. | |
| Yes | 1.1 (0.9, 1.3) | 0.536 |
| MRD% at end of Course 1 | | |
| <0.05 | Ref. | |
| ≥0.05 | 2.8 (2.3, 3.3) | <0.001 |
| MRD% at end of Course 2 | | |
| <0.05 | Ref. | |
| ≥0.05 | 3.7 (3.0, 4.6) | <0.001 |

**Abbreviations:** HL, Hyperleukopenia; WBC, white blood cell counts; PB blast, peripheral blood blast; BM, bone marrow blast; CNSL, central nervous system leukemia; CEBPA, CCAAT/enhancer binding protein alpha; FLT3-ITD, fms-related tyrosine kinase 3; NPM1, nucleophosmin 1; WT1, wilms tumor 1; CR, complete remission; MRD, minimal residual disease; SCT, stem cell transplantation.

**Table 3.** Net reclassification improvement (NRI) analysis comparing the new MRD threshold (0.05%) to the traditional threshold (0.1%) at end of course 1 based on multivariable model comparison. The statistical significance of the NRI was assessed using bootstrap resampling (with 1,000 iterations) to generate confidence intervals; *p*-values are two-sided.

| Group | Reclassification Improvement (%) | Reclassification Worsening (%) | NRI (95% *CI*) | *P*-value |
|---|---|---|---|---|
| Event Group | 15 | 5 | 0.10 (0.05–0.15) | 0.001 |
| Non-event Group | 10 | 5 | 0.05 (0.02–0.08) | 0.01 |
| Overall NRI | | | 0.15 (0.05–0.25) | 0.003 |

**Table 4.** Net reclassification improvement (NRI) analysis comparing the new MRD threshold (0.05%) to the traditional threshold (0.1%) at end of course 2 based on multivariable model comparison. The statistical significance of the NRI was assessed using bootstrap resampling (with 1,000 iterations) to generate confidence intervals; *p*-values are two-sided.

| Group | Reclassification Improvement (%) | Reclassification Worsening (%) | NRI (95% *CI*) | *P*-value |
|---|---|---|---|---|
| Event Group | 18.5 | 6.5 | 0.12 (0.06–0.18) | <0.001 |
| Non-event Group | 10.5 | 4.5 | 0.06 (0.02–0.10) | 0.005 |
| Overall NRI | | | 0.18 (0.08–0.28) | 0.001 |

**Table 5. Multivariable analysis for first event among pediatric AML.** *P*-values were derived from Wald tests; all *p*-values are two-sided.

| Outcome | Variable | *HR* (95% *CI*) | *P* value |
|---|---|---|---|
| First Event | WBC ≥ 50 × 10⁹/L | 1.0 (0.2, 3.9) | 0.979 |
| | FLT3/ITD mutation | 1.6 (0.6, 4.2) | 0.342 |
| | WT1 mutation | 1.8 (1.1, 3.2) | 0.029 |
| | SCT in 1st CR | 0.2 (0.1, 0.8) | 0.023 |
| | MRD ≥ 0.1% at end of Course 1 | 1.8 (1.1, 3.2) | 0.029 |
| | MRD ≥ 0.1% at end of Course 2 | 3.1 (1.2, 8.3) | 0.022 |
| | MRD ≥ 0.05% at end of Course 1 | 6.7 (1.8, 24) | 0.004 |
| | MRD ≥ 0.05% at end of Course 2 | 3.3 (1.3, 10) | 0.011 |

**Abbreviations:** WBC, white blood cell counts; CEBPA, CCAAT/enhancer binding protein alpha; FLT3-ITD, fms-related tyrosine kinase 3; NPM1, nucleophosmin 1; WT1, wilms tumor 1; MRD, minimal residual disease; SCT, stem cell transplantation.

consequential than a false-positive classification. To address this, we conducted a pre-specified sensitivity analysis that weighted sensitivity twice as heavily as specificity. Reassuringly, this clinical-priority-weighted analysis consistently identified 0.05% as the optimal threshold, thereby confirming the robustness of our primary finding and its applicability to clinical decision-making, where minimizing missed relapses is paramount.

Our findings not only align with but significantly extend the growing body of evidence from recent studies advocating for more sensitive MRD assessment in acute leukemias [10–14]. Although MFC techniques evolved over the study period, the core principles of high-sensitivity MRD detection (e.g., acquisition of high cell numbers, standardized panels) remained consistent within the COG reference laboratory, minimizing the impact of technological drift on threshold determination. The consistent optimal threshold identified across the entire cohort supports its robustness.

The central finding of this study—that an MRD threshold of 0.05% is prognostically optimal—resonates with the evolving paradigm in the field. Our ROC curve analysis, which identified 0.05% as the threshold with the maximum Youden's index for predicting 5-year EFS, provides robust statistical support for this threshold. This finding is consistent with several contemporary investigations. A 2023 study by Bazinet and colleagues [15]. on AML demonstrated that MRD levels below 0.1% but above 0.01% still carried significant prognostic value, suggesting the conventional threshold might be insufficiently sensitive. Similarly, in adult AML, recent work by Grob and colleagues (2023) utilizing next-generation sequencing approaches has advocated for lowering the threshold for MRD negativity, emphasizing that even low-level persistence of molecular disease correlates with inferior outcomes [16].

The enhanced discriminatory power of the 0.05% threshold was unequivocally demonstrated in our Kaplan–Meier survival analyses. The significantly higher *HRs* observed with the novel threshold underscore its superior ability to stratify patients based on relapse risk. This aligns with the previous findings which reported that a more sensitive threshold improved the prediction of relapse risk, particularly in patients with NPM1-mutated AML and CBF translocations who traditionally would have been classified as MRD-negative [17–19]. Our data provide robust, multivariable validated evidence that strengthens these emerging observations.

A critical contribution of our study is the quantitative demonstration of improved risk classification through NRI analysis. The significant NRI confirms that the 0.05% threshold moves patients into more appropriate risk categories, a metric that has gained prominence in recent prognostic model evaluations [20]. This addresses a key limitation of many previous studies that reported differences in *HRs* but did not quantify the improvement in classification accuracy. Our approach mirrors the rigorous methodology called for in a 2025 commentary Nachmias and colleagues [21]. on evaluating new prognostic biomarkers in oncology, ensuring our findings are clinically relevant beyond mere statistical significance.

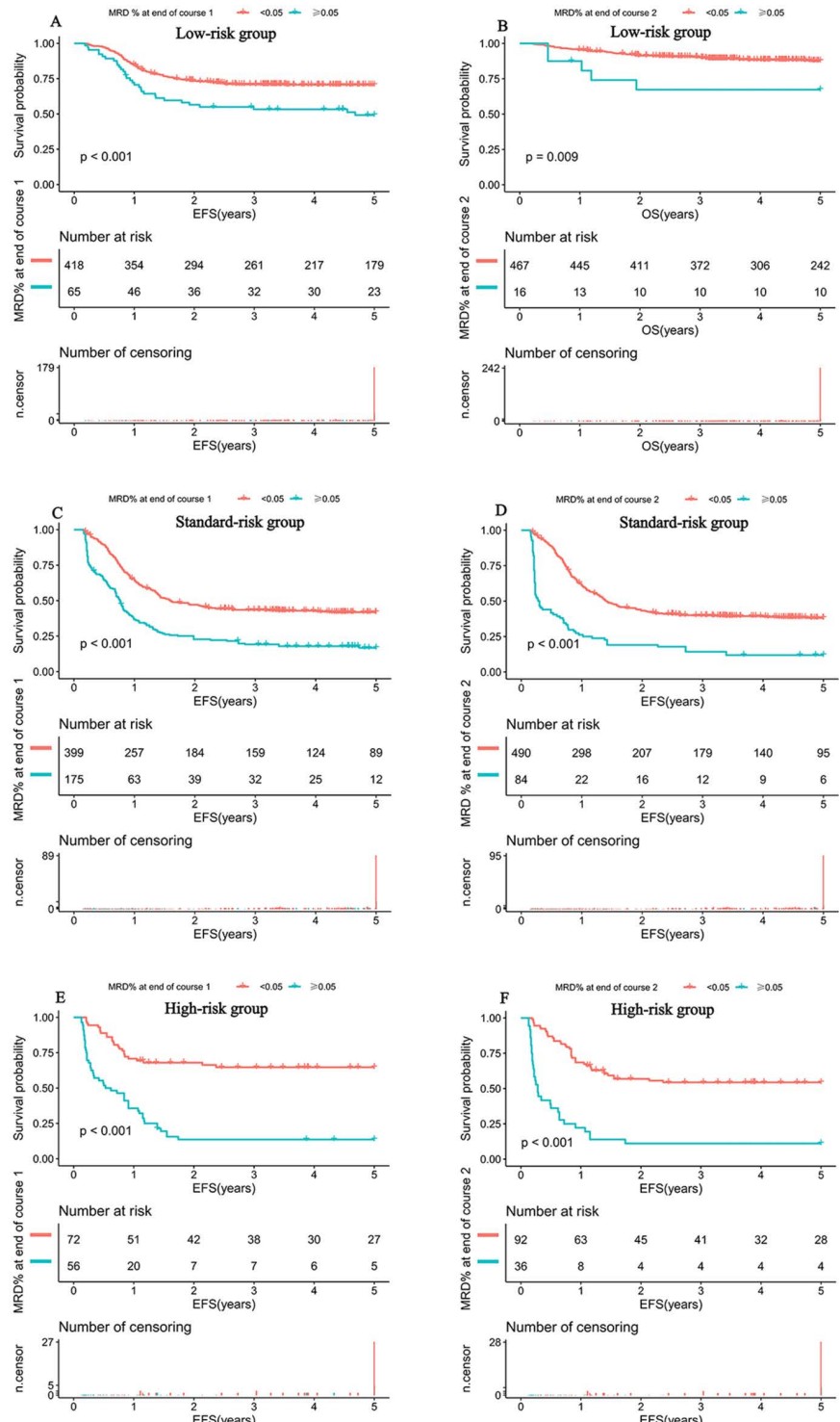

**Fig 4. Subgroup analysis of event-free survival (EFS) stratified by the novel MRD threshold of 0.05% within standard-risk (SR), intermediate-risk (IR), and high-risk (HR) groups after induction therapy in pediatric AML.** Survival curves were compared using the log-rank test; p-values are two-sided. **(A)** EFS for patients in the SR group, stratified by MRD status after the end of induction Course 1. **(B)** EFS for patients in the SR group, stratified by MRD status after the end of induction Course 2. **(C)** EFS for patients in the IR group, stratified by MRD status after the end of induction Course 1.

**(D)** EFS for patients in the IR group, stratified by MRD status after the end of induction Course 2. **(E)** EFS for patients in the HR group, stratified by MRD status after the end of induction Course 1. **(F)** EFS for patients in the HR group, stratified by MRD status after the end of induction Course 2.

Perhaps the most compelling evidence for the universal applicability of the 0.05% threshold is its consistent prognostic validity across all tested genetic risk subgroups. This finding is particularly noteworthy in the context of recent treatment advances. With the incorporation of targeted agents like FLT3 inhibitors and BCL-2 inhibitors into frontline pediatric trials, the depth of remission achieved with conventional chemotherapy is being redefined. Our data suggest that highly sensitive MRD monitoring will be crucial for assessing response to these novel regimens, as demonstrated in recent adult studies (Bataller and colleagues, 2020), their study revealed that the persistence of disease at the 0.05% level may identify patients who could benefit from early intervention or maintenance therapy. The multivariable analysis in this present study confirming MRD ≥0.05% as one of the strongest independent prognostic factors reinforces its fundamental role in risk stratification. This finding places MRD assessment squarely alongside, and perhaps above, established molecular markers in importance.

The adoption of a 0.05% MRD threshold has immediate and profound implications for clinical practice and trial design. It promises to refine patient selection for treatment intensification, such as allogeneic SCT in first remission. However, our findings suggest that the 0.05% MRD threshold may offer improved risk stratification compared to the 0.1% threshold, it is crucial to acknowledge the limitations inherent in our study design. The retrospective nature of data collection from multiple COG trials (e.g., AAML03P1, AAML0531, AAML1031) conducted over a 20-year period introduces heterogeneity in treatment regimens, supportive care, and technological evolution of flow cytometry. Although we applied standardized statistical adjustments, the amalgamation of these diverse cohorts may limit the direct extrapolation of our results to specific contemporary or future therapy protocols. Therefore, our findings should be interpreted as generating a compelling hypothesis rather than providing definitive evidence. We recommend that future prospective studies, particularly in uniformly treated cohorts using modern regimens (e.g., those incorporating FLT3 or BCL-2 inhibitors), validate the 0.05% threshold to confirm its generalizability.

Our study has limitations that should be acknowledged. Firstly, the retrospective design, though analyzing a large prospective cohort, necessitates validation in a purely prospective setting. Secondly, our analysis focused on key molecular subsets with the most complete data annotation within the cohort. While this ensures analytical robustness, it means that less frequent genetic aberrations or those with significant missing data were not included. Future studies with prospective, comprehensive genomic profiling will be valuable to further validate the prognostic performance of the 0.05% MRD threshold across the full spectrum of pediatric AML genetic subtypes. Finally, while our study focuses on a single time point threshold, future studies should investigate the prognostic impact of MRD kinetics, as a rising MRD level may be an even more critical indicator of impending relapse than persistent low-level disease.

In conclusion, this study establishes that a lower MRD threshold of 0.05%, measurable by standardized flow cytometry, is superior to the conventional 0.1% threshold for risk stratification in pediatric AML. Our findings, consistent with the direction of recent literature, argue for a paradigm shift in how we define treatment response. Adopting this more sensitive threshold may enable more precise risk-adapted therapy, ultimately improving outcomes for children with AML. We strongly advocate for its integration into future clinical trials and, upon validation, standard clinical practice.

## 4.1 Ethics approval and consent to participate

The original data collection within the COG trials was approved by the IRB of all participating centers, including the lead IRB at the National Cancer Institute, with written informed consent obtained from all participants or their guardians. The analysis of de-identified data for this specific study was reviewed and approved by the IRB of Sun Yat-sen Memorial Hospital (Approval No. SYSEC2-2025-BA-885).

## Supporting information

**S1 Checklist. STROBE checklist for observation studies.** This checklist is available under the Creative Commons Attribution 4.0 License (https://creativecommons.org/licenses/by/4.0/deed.en) and is adapted from the STROBE Statement (https://www.strobe-statement.org/).
(DOCX)

**S1 Code. R scripts for data analysis.**
(ZIP)

**S1 Protocol. Full study protocol and statistical analysis plan.**
(DOCX)

## Acknowledgments

The authors thank the Therapeutically Applicable Research to Generate Effective Treatments (TARGET) initiative, administered by the National Cancer Institute, for providing the data used in this analysis. The results published here are in whole or part based upon data generated by the TARGET initiative (https://ocg.cancer.gov/programs/target). We also acknowledge the patients and their families, as well as the participating institutions and researchers within the Children's Oncology Group, whose collective efforts made this resource possible.

## Author contributions

**Conceptualization:** Xiong-yu Liao, Hong Zheng, Jian-pei Fang, Kun-yin Qiu.

**Data curation:** Xiong-yu Liao, Hong Zheng, Jian-pei Fang, Kun-yin Qiu.

**Formal analysis:** Xiong-yu Liao, Hong Zheng.

**Investigation:** Dun-hua Zhou.

**Methodology:** Jian-pei Fang, Dun-hua Zhou.

**Project administration:** Jian-pei Fang.

**Software:** Hong Zheng.

**Validation:** Kun-yin Qiu.

**Visualization:** Kun-yin Qiu.

**Writing – original draft:** Xiong-yu Liao.

**Writing – review & editing:** Dun-hua Zhou, Kun-yin Qiu.

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
