## [Editor Report · Decision Letter 0]

2 Dec 2025

Dear Dr Qiu,

Thank you for submitting your manuscript entitled "Optimal Minimal Residual Disease Threshold in Pediatric Acute Myeloid Leukemia: A Prognostic Reevaluation Based on the TARGET Cohort" for consideration by PLOS Medicine.

Your manuscript has now been evaluated by the PLOS Medicine editorial staff, and I am writing to let you know that we would like to send your submission out for external peer review.

For clinical studies, please upload a copy of your trial study protocol as a supporting information file. The study protocol should be the version submitted for approval to the institutional review board or ethics committee, should include any amendments to the study protocol, as well as the date of their approval by the institutional review or ethics committee. Please also detail any deviations from the study protocol in the Methods section of your manuscript. The editors will consider the protocol and study conduct prior to a final decision for external review.

Please re-submit your manuscript within two working days, i.e. by Dec 04 2025 11:59PM.

Kind regards,

Heather Van Epps, PhD

Consulting Editor

PLOS Medicine

---

## [Decision Letter · Decision Letter 1]

16 Mar 2026

Dear Dr Qiu,

Many thanks for submitting your manuscript "Optimal Minimal Residual Disease Threshold in Pediatric Acute Myeloid Leukemia: A Prognostic Reevaluation Based on the TARGET Cohort" (PMEDICINE-D-25-04201R1) to PLOS Medicine. The paper has been reviewed by subject experts and a statistician; their comments are included below and can also be accessed here: [LINK]

As you will see, the reviewers had positive comments on the potential clinical relevance of your work, however several important methodological and clarity-related concerns have been raised. After discussing the paper with the editorial team and an academic editor with relevant expertise, I'm pleased to invite you to revise the paper in response to the reviewers' comments. We plan to send the revised paper to some or all of the original reviewers, and we cannot provide any guarantees at this stage regarding publication.

We ask that you submit your revision by Apr 06 2026 11:59PM. However, if this deadline is not feasible, please contact me by email, and we can discuss a suitable alternative.

Don't hesitate to contact me directly with any questions (efourli@plos.org).

Best regards,

Evangelia

Evangelia Fourli, Ph.D.

Associate Editor

PLOS Medicine

efourli@plos.org

Comments from the academic editor and editorial team:

- Please revisit your data availability statement to comply with PLOS Medicine requirements.

• If the data are held or will be held in a public repository, include URLs, accession numbers or DOIs. If this information will only be available after acceptance, indicate this by ticking the box below. For example: All XXX files are available from the XXX database (accession number(s) XXX, XXX.).

• If the data are all contained within the manuscript and/or Supporting Information files, enter the following: All relevant data are within the manuscript and its Supporting Information files.

• If neither of these applies but you are able to provide details of access elsewhere, with or without limitations,

please do so. For example:

Data cannot be shared publicly because of [XXX]. Data are available from the XXX Institutional Data Access / Ethics

Committee (contact via XXX) for researchers who meet the criteria for access to confidential data.

The data underlying the results presented in the study are available from (include the name of the third party and contact information or URL). This text is appropriate if the data are owned by a third party and authors do not have permission to share the data.

For more information visit: https://journals.plos.org/plosmedicine/s/data-availability

- Please revise your abstract section to comply with our requirements. The Abstract should include three sections: Background, Methods and Findings, and Conclusions. It should also provide all the information relevant to this study type. Please see https://journals.plos.org/plosmedicine/s/submission-guidelines#loc-abstract.

- In the abstract, please include the important dependent variables that are adjusted for in the analyses. Also, please report numbers (e.g. the # of patients whose data was analysed), and the results of the analyses, using 95% CIs where applicable. The Abstract should include more details on the study question and cohort for a general reader.

- The last paragraph of your introduction should be restructured to include the main aims of the study.

- Please specify the institutional review boards in section 2.1 and the type of consent obtained, ie written or oral.

- Section 2.2, you mention 4 protocols but you reference only 3. Please, correct as appropriate.

- Figures 1, 3 and 4, the numbering on the y axis of some panels is overlapping close to 0. Please correct this if possible.

- Please include the statistical tests used for p-value calculation in all relevant figure and table legends.

Comments from the reviewers:

Reviewer #1: This is cohort study that compared the threshold for minimal residual disease (MRD) for paediatric acute myeloid leukaemia (AML) patients. It compared the traditional (0.1%) with a newly proposed threshold (0.05%).

Major:

- Multivariable analysis selected variables based on arbitrary p-value threshold (<0.1) which is not optimised for prediction performance. Please consider either AIC or LASSO method for variable selection, both of which optimises prediction performance. Please also make sure the aim of the multivariate analysis is clear - right now it does not indicate whether it is for prognostic or explanatory research even though implied from the research question.

- Also because of how the variables were selected for multivariable analysis, both MRD >= 0.1% and MRD >= 0.05% were included in the model which are hard to interpret (there were overlaps in the categories). Please consider creating a categorical variable of 3 levels: MRD < 0.05%; 0.05 - <0.1%; >=0.01% so that dose response associations are clearer.

- Also even though a multivariable model was developed, it was not used in the AUC / NRI analysis. This also leads to the question on why the multivariable model was developed.

- Why Youden's index was used to determine 'optimal' threshold? Youden's index assign equal weighting for both false positive and false negative which surely won't be the case in this clinical setting?

- Can you please stratify Table 1 by the outcome status (the composite outcome) so that it is clearer the absolute differences?

Minor:

- '0.05% threshold demonstrated higher hazard ratios for EFS' - 'EFS' should be 'composite outcome' or something similar. HR is a statistic for an event, not survival.

- 'multivariate' should be 'multivariable'. The former indicates multiple outcomes. Same in the main text.

- 'HR' is used both for 'hazard ratio' and 'high risk' which is confusing.

Reviewer #2: The authors have analyzed data from the TARGET-AML cohort regarding the additional value (if any) of a lower (0.05%) than conventional (0.1%) treshhold for flow cytometry based MRD positivity or negativity. They conclude that the lower treshhold performs better in predicting outcome. This is an important finding and should stimulate consortia to improve their flow cytometry methods to enable higher sensitivity of the assay.

Some comments to further improve this manuscripts:

1. Introduction: in stating that MFC-MRD is among the most powerful predictors of outcome, they refer to a paper on molecular monitoring. More appropriate would be the review by Segerink et al., in Exp Rev Anticancer Ther 2021. Along this line, reference #8 also does not seeem to concern MFC-based MRD monitoring.

2. Only 4 molecularly defined subsets were defined, but there are many more. Why not include more subsets in the analyses?

3. Page 5 at the top: four protocols but only 3 are mentioned ......

4. What is there was no LAIP? Different from normal?

5. Page 6: the authors refer to established cytogenetic and molecular criteria. Please add a reference! Perhaps the recent consensus paper in BLOOD?

6. It must be made more clear when induction course 2 was started in reality. In other words, starting criteria for course 2. This is important, since it may impact MRD levels. Especially when using 'intensified' induction chemotherapy, induction course 2 will start as soon as possible in poor responders. Was this concept used in these COG trials? It is pivotal to know whether MRD levels impacted the induction chemotherapy course 2.

7. Page 7: median and range of MRD levels are mentioned, but also mention the percentage of MRD positivity after course 1 and 2 when using both treshholds (0.1 and 0.05%).

8. Page 8: sensitivity and specificity for the cut-off of 0.1% must also be provided, both for after course 1 and course 2.

9. In the discussion, please emphasize that not so much a low level of persistint MRD is important, but a rising MRD.

10. Page 11: please do not mention individual drugs, the type of drugs is enough to illustrate it.

11. What remains is in which percentage of children it was possible to determine a level below 0.1%. That is somewhat hidden, please make that more clear in the results and discussion sections.

Finally, as detail: sex instead of gender?

Reviewer #3: Liao and coworkers describe a retrospective analysis of FLOW-MRD data from the NCI TARGET cohort of pediatric patients (n= 1205) treated for AML between 1996 and 2016 upon COG trials. They focus on determining the optimal MRD threshold after first as well as second induction for outcome prediction measured by EFS. They use ROC curve analysis with maximum Youden's index determination of the optimal threshold and validate their findings with survival curve comparisons, multivariable analysis and especially NRI analysis in order to quantify predictive performance. The main finding of the study is that a threshold of 0.05% at the tested time-points performs better than the conventional 0.1% threshold.

Major critical issues:

1) The main draw-back of this technically well performed study is the retrospective nature of patient recruitment from several COG trials (at least 3, although authors claim 4: which is the fourth trial?) which evaluated different treatment and stratification criteria in the long period of 20 years of recruitment. It is nowadays well recognized that threshold levels and their usefulness are related to regimen before as well as after the time-point of analysis which is why a lump-sum analysis of several regimens brought together does not allow extrapolation of the meaning of a certain threshold for other and especially current or future regimens. Authors need to limit their enthusiasm regarding their findings as declared in the discussion towards a more realistic interpretation and rather recommend their methodology to researchers validating data from more uniform trial environments and treatment.

2) MFC methodology is not described and no references given. This is a must to understand the quality of the technology under investigation which might have changed and improved over time. Would such technological improvements have changed the results determined during sections of the long observation period?

3) Although EFS is the overarching prognostic correlate, MRD is more associated with response and relapse than with any other type of event. It is therefore necessary to run all analyses for relapse free survival in order to determine whether old or new thresholds are different in predicting relapse.

4) In 2.2, stratification criteria are described, with high-risk (HR) including only monosomy 7 or del(5q). In Table 1 the number of HR patients is 128 which makes up for 10.8% of all patients. This is hardly credible. Whereas nowadays HR patients are defined by a much larger selection of genetic aberrations (and/or poor response), it is simply impossible by incidence in pediatric AML that monosomy 7 and del(5q) render a high risk group of 10%. This needs explanation. This obvious flaw also impacts the finding of a universal applicability of the 0.05% threshold across all risk groups. Risk grouping should be redone upon current criteria (see Zwaan et al. Blood. 2025 Oct 17:blood.2024027904. doi: 10.1182/blood.2024027904).

Minor critical issues:

1) language flaws: asensitivity (Introduction); nevo-AML >> de novo-AML (chapter 2.1 and 3.1)

2) "MLL" in Table 1 and 2 should be changed for the current terminology "KMT2A"

3) Figure 1 legend; described is the "old" threshold at 0.1%, however, the legend reads ...using the new threshold of 0.1"... (line 3 as well as 4) >> delete "new"

4) Figure 4: Plot A & B show headings "Standard risk group": this is not correct because these curves are (by numbers and outcome) representing the "Low risk group". Vice-versa, plots C & D are headed by "Low risk group" which is false and should be changed for "Standard risk group".

---

* Please upload any figures associated with your paper as individual TIF or EPS files with 300dpi resolution at resubmission; please read our figure guidelines for more information on our requirements: http://journals.plos.org/plosmedicine/s/figures. While revising your submission, we strongly recommend that you use PLOS's NAAS tool (https://ngplosjournals.pagemajik.ai/artanalysis) to test your figure files. NAAS can convert your figure files to the TIFF file type and meet basic requirements (such as print size, resolution), or provide you with a report on issues that do not meet our requirements and that NAAS cannot fix.

After uploading your figures to PLOS's NAAS tool - https://ngplosjournals.pagemajik.ai/artanalysis, NAAS will process the files provided and display the results in the "Uploaded Files" section of the page as the processing is complete.

If the uploaded figures meet our requirements (or NAAS is able to fix the files to meet our requirements), the figure will be marked as "fixed" above. If NAAS is unable to fix the files, a red "failed" label will appear above.

When NAAS has confirmed that the figure files meet our requirements, please download the file via the download option, and include these NAAS processed figure files when submitting your revised manuscript.

* Please ensure that the study is reported according to the STROBE or REMARK guideline and include the completed checklist as Supporting Information. When completing the checklist, please use section and paragraph numbers, rather than page numbers. Please add the following statement, or similar, to the Methods: "This study is reported as per STROBE or REMARK guideline (S1 Checklist)."

FIGURES AND TABLES

SUPPLEMENTARY MATERIAL

REFERENCES

OBSERVATIONAL STUDIES

* Abstract: Please include the study design, population and setting, number of participants, years during which the study took place (enrollment and follow up), length of follow up, and main outcome measures.

* Please ensure that the study is reported according to the STROBE (or appropriate STOBE extension) guideline (available from: https://www.equator-network.org/reporting-guidelines/strobe) and include the completed STROBE (or STROBE extension) checklist as Supporting Information. Please add the following statement, or similar, to the Methods: "This study is reported as per the Strengthening the Reporting of Observational Studies in Epidemiology (STROBE) guideline (S1 Checklist)." When completing the checklist, please use section and paragraph numbers, rather than page numbers. For your type of study you could choose to report according to the REMARK checklist.

* [FOR POPULATION HEALTH/REGISTRY STUDIES] Please ensure that the study is reported according to the RECORD guideline (available from https://www.record-statement.org) and include the completed checklist as Supporting Information. Please add the following statement, or similar, to the Methods: "This study is reported as per the Reporting of Studies Conducted using Observational Routinely-Collected Data (RECORD) guideline (S1 Checklist)." When completing the checklist, please use section and paragraph numbers, rather than page numbers.

* [FOR POPULATION HEALTH ESTIMATES] Please ensure that the study is reported according to the GATHER statement (available from https://www.equator-network.org/reporting-guidelines/gather-statement) and include the completed checklist as Supporting Information. Please add the following statement, or similar, to the Methods: "This study is reported as per the Guidelines for Accurate and Transparent Health Estimates Reporting (GATHER) statement (S1 Checklist)." When completing the checklist, please use section and paragraph numbers, rather than page numbers.

* [FOR MEDIATION ANALYSES] We recommend that the study is reported according to the AGReMA statement (https://agrema-statement.org/#:~:text=AGReMA%20is%20an%20evidence%2D%20and,randomised%20trials%20and%20observational%20studies) and include the completed checklist as Supporting Information. Please add the following statement, or similar, to the Methods: "This study is reported as per the Guideline for Reporting Mediation Analyses (AGReMA) statement (S1 Checklist)." When completing the checklist, please use section and paragraph numbers, rather than page numbers.

* For all observational studies, in the manuscript text, please indicate: (1) the specific hypotheses you intended to test, (2) the analytical methods by which you planned to test them, (3) the analyses you actually performed, and (4) when reported analyses differ from those that were planned, transparent explanations for differences that affect the reliability of the study's results. If a reported analysis was performed based on an interesting but unanticipated pattern in the data, please be clear that the analysis was data driven.

* Please state in the Methods section whether the study had a prospective protocol or analysis plan. If a prospective analysis plan (from your funding proposal, IRB or other ethics committee submission, study protocol, or other planning document written before analyzing the data) was used in designing the study, please include the relevant document(s) with your revised manuscript as a Supporting Information file to be published alongside your study and cite it in the Methods section. A legend for this file should be included at the end of your manuscript. If no such document exists, please make sure that the Methods section transparently describes when analyses were planned, and when/why any data-driven changes to analyses took place. Changes in the analysis, including those made in response to peer review comments, should be identified as such in the Methods section of the paper, with rationale.

---

## [Decision Letter · Decision Letter 2]

15 Apr 2026

Dear Dr. Qiu,

Thank you very much for re-submitting your manuscript "Optimal Minimal Residual Disease Threshold in Pediatric Acute Myeloid Leukemia: A Prognostic Reevaluation Based on the TARGET Cohort" (PMEDICINE-D-25-04201R2) for review by PLOS Medicine.

I have discussed the paper with my colleagues and the academic editor and it was also seen again by 3 reviewers. One of the reviewers has requested some wording changes, mainly to more accurately reflect the inherent limitations of the data.

I am pleased to say that provided you address the reviewer's requests and the remaining editorial and production issues are dealt with we are planning to accept the paper for publication in the journal.

[LINK]

We look forward to receiving the revised manuscript by Apr 22 2026 11:59PM.

Sincerely,

Evangelia Fourli, Ph.D.

Senior Editor

PLOS Medicine

plosmedicine.org

Requests from Editors:

Please note that some of the following requests may not apply to your manuscript.

GENERAL EDITORIAL REQUESTS

"* At this stage, we ask that you include a short, non-technical Author Summary of your research to make findings accessible to a wide audience that includes both scientists and non-scientists. The Author Summary should immediately follow the Abstract in your revised manuscript. This text is subject to editorial change and should be distinct from the scientific abstract. Ideally each sub-heading should contain 2-3 single sentence, concise bullet points containing the most salient points from your study. In the final bullet point of ‘What Do These Findings Mean?’ Please include the main limitations of the study in non-technical language.

Please see our author guidelines for more information: https://journals.plos.org/plosmedicine/s/revising-your-manuscript#loc-author-summary."

* Please confirm that your title complies with PLOS Medicine's style. Your title must be nondeclarative and not a question. It should begin with main concept if possible. "Effect of" should be used only if causality can be inferred, i.e., for an RCT. Please place the study design ("A randomized controlled trial," "A retrospective study," "A modelling study," etc.) in the subtitle (ie, after a colon).

* Please confirm that your abstract complies with our requirements, including format (three sections: Background, Methods and Findings, and Conclusions) and providing all the information relevant to this study type https://journals.plos.org/plosmedicine/s/submission-guidelines#loc-abstract

* Please ensure that the Introduction ends with a clear description of the study question or hypothesis.

* Please ensure that all abbreviations are defined at first use throughout the text.

* Please confirm that all numbers presented in the abstract are present and identical to numbers presented in the main manuscript text.

GENERAL

* Please review your text for claims of novelty or primacy (e.g. 'for the first time') and remove this language. In addition, please check that any use of statistical terms (such as trend or significant) are supported by the data, and if not please remove them.

* Please remove the 'conclusions' subheading from the discussion. Please also remove any other subheadings from the discussion.

"* Statistical reporting: Please revise throughout the manuscript, including tables and figures.

- Please report statistical information as follows to improve clarity for the reader ""22% (95% CI [13,28]; p</=)"".

- Please separate upper and lower bounds with commas instead of hyphens as the latter can be confused with reporting of negative values.

- Please repeat statistical definitions (HR, CI etc.) for each set of parentheses."

* In the author summary, please revise formatting and ensure you use bullet points.

* Paragraph 2.4.3 Analyses Performed: please add bullet points or a type of numbering to each analysis described.

*Line 228, remove "A" and substitute "was" with "were"

*Line 273, please insert a space between " not)." and "The". Same for line 319 "subgroup.Applying".

*Please revise S1 Checklist for punctuation errors.

* Your study is observational and therefore causality cannot be inferred. Please remove language that implies causality and refer to associations instead.

FUNDING STATEMENT

* The funding statement should include: specific grant numbers, initials of authors who received each award, URLs to sponsors’ websites. Also, please state whether any sponsors or funders (other than the named authors) played any role in study design, data collection and analysis, the decision to publish, or preparation of the manuscript. If they had no role in the research, include this sentence: “The funders had no role in study design, data collection and analysis, decision to publish, or preparation of the manuscript.”

COMPETING INTERESTS STATEMENT

* All authors must declare their relevant competing interests per the PLOS policy, which can be seen here: https://journals.plos.org/plosmedicine/s/competing-interests For authors with ties to industry, please indicate whether any of the interests has a financial stake in the results of the current study.

DATA AVAILABILITY

"* PLOS Medicine requires that the de-identified data underlying the specific results in a published article be made available, without restrictions on access, in a public repository or as Supporting Information at the time of article publication, provided it is legal and ethical to do so. Please see the policy at

http://journals.plos.org/plosmedicine/s/data-availability

and FAQs at

http://journals.plos.org/plosmedicine/s/data-availability#loc-faqs-for-data-policy "

"* The Data Availability Statement (DAS) requires revision. For each data source used in your study:

"

ETHICS AND CONSENT

*Please clarify in the text whether informed consent was waived due to the use of de identified data.

* "The original data collection within the Children’s Oncology Group (COG) trials was approved by the Institutional Review Boards (IRB) of all participating centers, including the lead IRB at the National Cancer Institute, with written informed consent obtained from all participants or their guardians." Please cite the original paper reporting those IRBs in the ethics statement in the Methods.

FIGURES

* Please show graph axes beginning at zero. If this is not possible, please show a break in the axis.

* In the Kaplan-Meier curve(s) please provide the number at risk for each time interval.

* When a p value is given, please specify the statistical test used to determine it in the legend.

Acknowledgments

* Please include an Acknowledgments section in your manuscript.

* Acknowledgments must not include funders and/or authors of the study.

Comments from Reviewers:

Reviewer #1: Thank you for addressing my comments. I appreciate the additional analyses and clarification. I have no further comments. Congratulations on completing this study.

Reviewer #2: The revision is fine with me, all my comments have been addressed appropriately.

Reviewer #3: The revised version includes thorough adaptations, however, few further changes appear essential to fulfill the tenor of criticism. Essentially, this analysis depends on COG data from a long period of recruitment, their validity for other and contemporary treatment environments can only be supposed but not taken for granted.

Change wording

1) in the abstract lines 49-52 - new wording:

Conclusions A lower MRD threshold of 0.05% provides superior prognostic discrimination compared to the conventional 0.1% threshold in pediatric AML treated in previous COG trials. These findings support testing this more sensitive threshold in future clinical trial designs for improved risk-adapted therapy.

2) in the discussion, p.14, lines 466 - 467 - new wording:

Perhaps the most compelling evidence for the universal applicability of the 0.05% threshold is its consistent prognostic validity across all tested genetic risk subgroups.

3) in the discussion, p.15, lines 499 - 501 - new wording:

While this ensures analytical robustness, it means that less frequent genetic aberrations or those with significant missing data were not included.

4) in the conclusion, p.16, lines 514 - 515 - new wording:

Adopting this more sensitive threshold may enable more precise risk-adapted therapy, ultimately improving outcomes for children with AML.

[LINK]

---

## [Editor Report · Decision Letter 3]

21 Apr 2026

Dear Dr Qiu,

On behalf of my colleagues and the Academic Editor, Dr Paschalis Evangelidis, I am pleased to inform you that we have agreed to publish your manuscript "Optimal Minimal Residual Disease Threshold in Pediatric Acute Myeloid Leukemia: A Retrospective Cohort Study Based on the TARGET Database" (PMEDICINE-D-25-04201R3) in PLOS Medicine.

Before your manuscript can be formally accepted you will need to complete some formatting changes and final editor requests, which you will receive in a follow up email. Please be aware that it may take several days for you to receive this email; during this time no action is required by you. Once you have received these formatting requests, please note that your manuscript will not be scheduled for publication until you have made the required changes.

PRESS

Sincerely,

Evangelia Fourli, Ph.D.

Senior Editor

PLOS Medicine